# Single synapse evaluation of the postsynaptic NMDA receptors targeted by evoked and spontaneous neurotransmission

Austin L Reese[1], Ege T Kavalali[2]*

[1]Departments of Neuroscience, UT Southwestern Medical Center, Dallas, United States; [2]Departments of Physiology, UT Southwestern Medical Center, Dallas, United States

**Abstract** Recent studies indicate that within individual synapses spontaneous and evoked release processes are segregated and regulated independently. In the hippocampus, earlier electrophysiological recordings suggested that spontaneous and evoked glutamate release can activate separate groups of postsynaptic NMDA receptors with limited overlap. However, it is still unclear how this separation of NMDA receptors is distributed across individual synapses. In a previous paper (*Reese and Kavalali, 2015*) we showed that NMDA receptor mediated spontaneous transmission signals to the postsynaptic protein translation machinery through $Ca^{2+}$-induced $Ca^{2+}$ release. Here, we show that in rat hippocampal neurons although spontaneous and evoked glutamate release driven NMDA receptor mediated Ca2+ transients often occur at the same synapse, these two signals do not show significant correlation or cross talk.

*For correspondence: ege.
kavalali@utsouthwestern.edu

**Competing interests:** The authors declare that no competing interests exist.

## Introduction

The vast majority of neuronal subtypes in the mammalian central nervous system exhibit both action potential (AP) synchronized evoked neurotransmitter release and stochastic, spontaneous neurotransmitter release (*Kavalali, 2015*). Previous studies by our group and others have shown that spontaneous neurotransmission is critical in the homeostatic regulation of synaptic strength and this regulation may have implications for the efficacy of rapidly acting antidepressants (*Nosyreva et al., 2013*; *Gideons et al., 2014*; *Sutton et al., 2006*). In particular, our recent work has shown that the $Ca^{2+}$ dependence of these effects is driven by NMDA receptor mediated $Ca^{2+}$ signals that are amplified via $Ca^{2+}$-induced $Ca^{2+}$ release via ryanodine receptor mediated coupling to postsynaptic $Ca^{2+}$ stores (*Reese and Kavalali, 2015*). These earlier results highlight a novel mechanism by which spontaneous release can trigger large $Ca^{2+}$ transients — and in turn activates postsynaptic signaling — but it remains unclear if these transients are generated at the same synapses that participate in evoked release.

An early investigation into the possibility that evoked and spontaneous neurotransmission may be separated spatially found that glutamate from evoked neurotransmitter release fell upon a subset of NMDA receptors distinct from those activated by spontaneous neurotransmission (*Atasoy et al., 2008*). However, these results lack the ability to distinguish quantal events at the single synapse level and do not show the extent to which evoked and spontaneous release processes are co-localized. More recent experiments in the *Drosophila* neuromuscular junction have come to mixed conclusions about the propensity of a single active zone to participate in both evoked and spontaneous transmission; finding either considerable overlap, but with a population of spontaneous only active zones

or limited overlap with a negative correlation between the two modes of transmission (*Melom et al., 2013*; *Peled et al., 2014*). To date, no such measurement has been made in mammalian synapses. Here, we utilize the genetically encoded $Ca^{2+}$ sensor GCaMP6f-PSD95 to image neurotransmission at individual synaptic regions to measure the amount of transmission mode overlap in hippocampal neurons.

## Results and discussion

### Visualization of AP5 sensitive postsynaptic $Ca^{2+}$ transients with GCaMP6f-PSD95

In order to detect $Ca^{2+}$ signals in the postsynaptic compartment, we utilized the postsynaptically targeted indicator GCaMP6f-PSD95. Dissociated hippocampal cultures DIV 14–16 were transfected using Lipofectamine 3000 (Thermo Fisher, Waltham MA) to produce sparse cell labeling (<1% efficiency). The indicator was expressed in the postsynaptic compartment where its clustering produced punctate signals consistent with a postsynaptic localization (*Figure 1A,B*). ROIs (regions of interest) were placed systematically over individual fluorescent puncta so as to minimize the chance of an ROI receiving signals from multiple postsynaptic fluorescence signals. Fluorescence traces were generated from ROIs and peaks corresponding to synaptic activity were counted such that one ROI correlates to one synapse (*Figure 1A*, see Materials and methods). To determine if the observed transients were in fact NMDA receptor mediated currents, we recorded cells for 4 min in Tyrode's solution containing 1 µM TTX and then TTX + 50 µM AP5. The addition of AP5 produced a 98.3% reduction in detected events over 4 min which is consistent with the recorded signals originating from postsynaptic NMDA currents (*Figure 1C–D*).

### Spontaneous response rate and evoked response probability do not correlate at the single synapse level

When subjected to field stimulation, individual puncta responded in a sparse, synchronous manner consistent with the evoked release of synaptic vesicles. To allow recording of both evoked and spontaneous responses, cells were perfused with Tyrode's solution containing 5 µM NBQX, 5 µM muscimol and 20 µM ryanodine. In this system, the AMPA receptor antagonist NBQX and the $GABA_a$ agonist muscimol function to prevent spontaneous action potentials and reverberatory network activity during stimulation by shunting excitation in the dendrite and soma but not the axon (*Figure 2—figure supplement 1A,B*) (*Sperk et al., 1997*). Ryanodine was included to block efflux of $Ca^{2+}$ from the endoplasmic reticulum which we have previously shown to contribute to synaptic $Ca^{2+}$ measurements (*Reese and Kavalali, 2015*).

Under these conditions we detected spontaneous and evoked response rates consistent with earlier estimates of evoked and spontaneous release probabilities at single synapses (*Murthy et al., 1997*; *Leitz and Kavalali, 2011*, *2014*). To assess if spontaneous response rate and evoked response probability (reported here as $R_p$ due to the postsynaptic nature of our measurement and so as not to be confused with presynaptic $P_r$) may be functionally correlated, cells were recorded for 8 min during which single action potentials were stimulated every 30 s for a total of 15 action potentials (see example *Video 1*, *Figure 2A*). Signal peaks detected within 1 s of a stimulus were considered to be evoked, and all peaks falling out of the 1 s window were considered to be spontaneous in origin (see Methods). When signal peak times are plotted, synchronous evoked activity across multiple ROIs is apparent on a background of spontaneous activity (*Figure 2A*).

Plotting the spontaneous response rate (*Figure 2B*) against the evoked response rate for each synapse reveals no linear correlation between the two parameters (*Figure 2C*). A fit line describes a trend towards lower spontaneous response rates in high evoked $R_p$ ROIs, but with very weak correlation. Interestingly, there is a large population of ROIs (~20%) that exhibit spontaneous but not evoked responses during the recording period. These ROIs may represent synapses that exclusively release vesicles spontaneously, or have a very low evoked $R_p$ (*Figure 2C*, along y axis). The presence of these ROIs suggests that a population of synapses may exist which exclusively signal spontaneously rather than by action potential driven chemical neurotransmission. The distribution of observed frequencies for spontaneous responses shows clustering around a mean rate of 0.78 events / ROI / minute (*Figure 2B*). Evoked responses show a more even distribution with a mean $R_p$ of 0.34

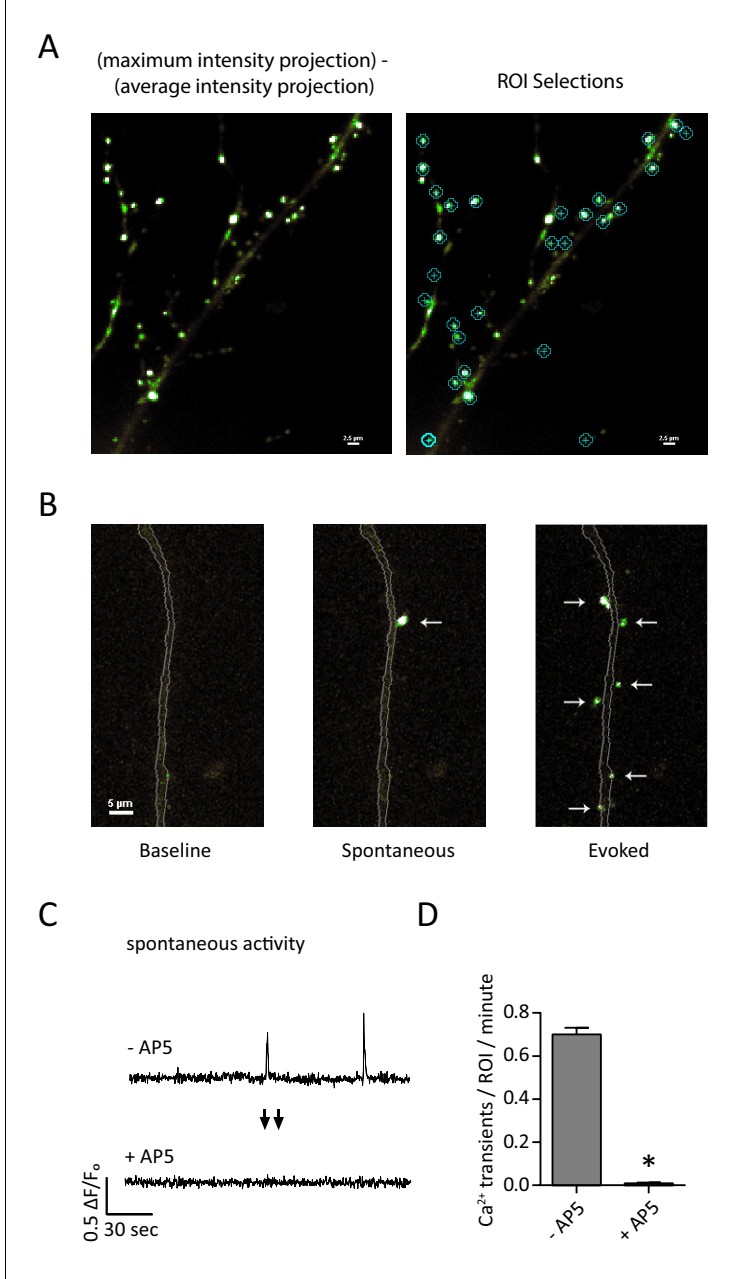

**Figure 1.** Detection of postsynaptic Ca²⁺ transients with GCaMP6f-PSD95. (**A**) Example images and ROI selection. Each image is a maximum intensity projection – average intensity projection for an 8 min recording both with (right) and without (left) the ROI selections. (**B**) Example images showing a section of dendrite (gray outline) during baseline fluorescence, peak fluorescence of a spontaneous event, and peak fluorescence of stimulus evoked events. Arrows indicate postsynaptic signals. (**C**) Example traces from a GCamp6f-PSD95 puncta before and after AP5 treatment. (**D**) Quantification showing Ca²⁺ transients detected in TTX before and after the addition of AP5. N = 600 ROIs from six experiments, two cultures, p<0.001 via students paired T-test.

(*Figure 2D*). In addition, we found that neither evoked $R_p$ or spontaneous response rate vary with respect to their distance to the soma (data not shown). Further analysis of this data was performed in which the coefficient of variance was calculated for the event amplitudes recorded at each ROI. These calculations reveal no correlation between spontaneous or evoked response rate and the

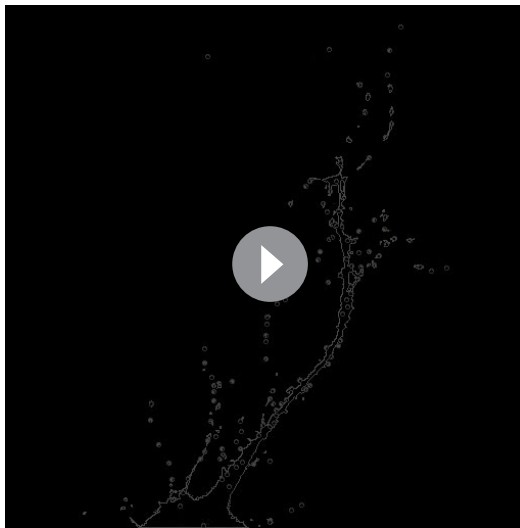

**Video 1.** Monitoring spontaneous and evoked neurotransmission on to single hippocampal neurons. To visualize spontaneous and evoked neurotransmission GCaMP6f-PSD95 transfected cells were recorded for 8 min during which single action potentials were stimulated every 30 s for a total of 15 action potentials.

variance in amplitudes that would be indicative of multiple synaptic inputs per ROI (*Figure 2—figure supplement 1C,D*).

Previous studies have shown that the distribution of spontaneous synaptic responses in time is not consistent with the events being independent at the population or single-synapse level and thus do not follow the Poisson distribution (*Abenavoli et al., 2002*; *Melom et al., 2013*). In our data, neither the Poisson distribution matching the measured average ($\lambda$ = 6.2, p<<0.001, excluding zeros, not shown) or the theoretical best fit Poisson distribution ($\lambda$ = 4.3, p<<0.001, excluding zeros) adequately describe the measured distribution as compared by the $\chi^2$ test (*Figure 2E*). We therefore conclude that the occurrence of spontaneous transmission events is a non-Poisson process and not truly random at the single synapse level.

## Use-dependent NMDAR block reveals sub-synaptic separation of evoked and spontaneous signals

To investigate whether evoked and spontaneously released glutamate activate common or separate pools of NMDARs within a single synapse, we utilized the use-dependent NMDAR blocker MK-801(*Huettner and Bean, 1988*). Cells were given 10 single stimuli, 10 s apart to establish a baseline $R_p$ before washing in 10 µM MK-801 for 10 min. After treatment and while MK-801 was still in the bath solution, cells were again stimulated 10 times, 10 s apart to measure $R_p$ after treatment (*Figure 3A*, bottom trace). To account for photobleaching and cellular run-down during this long recording period, control experiments were performed without MK-801 (*Figure 3A*, top). During the 10 min of stimulation-free MK-801 treatment, spontaneous response rate decreased by 89% which is consistent with MK-801 induced block of NMDARs that respond to spontaneous glutamate release (*Figure 3A,B*) (p=0.010 comparing first and last minute of MK-801 with student's paired t-test). In contrast, there was no significant decrease in spontaneous response rate in the control experiments when comparing the first and last minute (p=0.20, student's paired t-test) (*Figure 3B*).

In control cells, of the ROIs that responded one or more times during the first stimulation, spontaneous responses were observed in 75% of those ROIs and 58% responded to stimulation after the 10 min control perfusion. In the MK-801 treated cells, of the ROIs that responded during the first stimulation, 66% responded spontaneously and 56% responded after treatment (*Figure 3C*, blue bars). We expect that if MK-801 application at rest blocked a large number of both spontaneous and evoked release responding NMDA receptors, there would be a steep decrease in the number of synapses that respond to evoked stimulus after MK-801 treatment. However, the ability of ROIs to respond one or more times to evoked stimulus after MK-801 treatment is unaffected compared to control (*Figure 3C*, p=0.31 via one-way ANOVA). The application of MK-801 did not affect the number of spontaneously active ROIs (*Figure 3C*), though it did significantly decrease the rate of detected events at those ROIs (*Figure 3B*). While the vast majority of ROIs exhibit both spontaneous and evoked responses (*Figure 2C*, spontaneous bars), the inability of MK-801 to halt evoked responses after spontaneous block suggests a separation of the two modes of transmission within the same ROI.

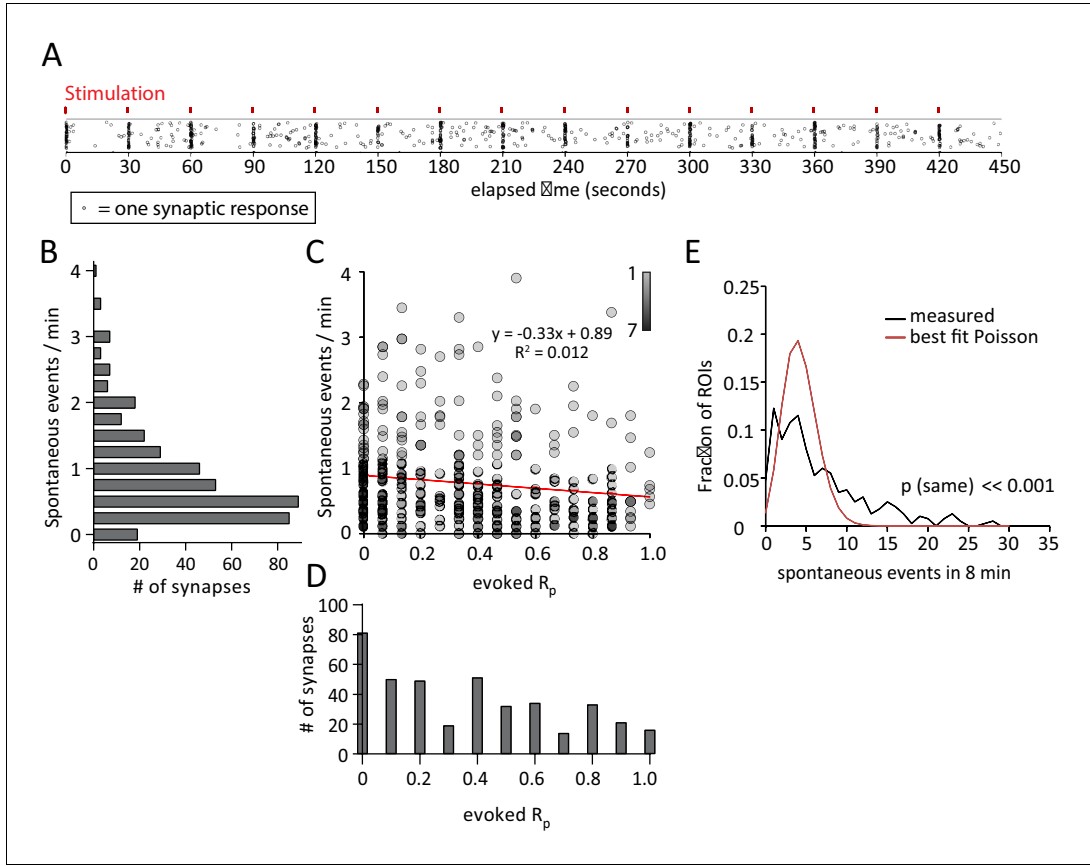

**Figure 2.** Spontaneous response frequency and evoked response probability show little correlation within the synapse. (**A**) Example results from an experiment where each mark is a single response from one of the 50 ROI's. Marks are scattered randomly in the y axis to facilitate visualization of responses that overlap in time. (**B**) Histogram showing spontaneous event rates per minute for 400 ROIs from 8 cells and four cultures. (**C**) Scatter plot of evoked release probability ($R_p$) vs spontaneous response rate for the same 400 ROIs. (**D**) Histogram showing the distribution of evoked response probabilities. (**A**) Distribution of ROIs by events per 8 min plotted with Poisson theoretical best fit. $\chi^2$ test indicates these are not equivalent distributions.

The following figure supplement is available for figure 2:

**Figure supplement 1.** NBXQ and muscimol control network activity while allowing field stimulation; $C_v$ analysis evidence that each ROI isolates a single synapse.

## High spontaneous response rate does not predict magnitude of evoked MK-801 block

During stimulation there was a decrease in amplitude with every subsequent response in the MK-801 treated cells and no single ROI responded more than eight times with MK-801 in the bath solution (*Figure 3D*). This decrease in amplitude is accompanied by a decrease in $R_p$ after MK-801 treatment that preferentially affects high responding ROIs which is consistent with its usage-dependent mechanism of action (*Figure 3E*). Channels that open during the spontaneous-only period of MK-801 treatment will be blocked and unavailable to participate in the second round of evoked responses if they were to bind glutamate.

Comparing the spontaneous event rate measured *before* the addition of MK-801 against the percent reduction in evoked $R_p$ after MK-801 treatment shows little correlation between the two parameters (*Figure 4A*). If NMDARs were equally likely to receive neurotransmitter from a spontaneous vesicle fusion as an evoked vesicle fusion, we would expect that the ROIs with the highest rate of spontaneous activity would exhibit the largest amount of MK-801 induced blockade when measuring

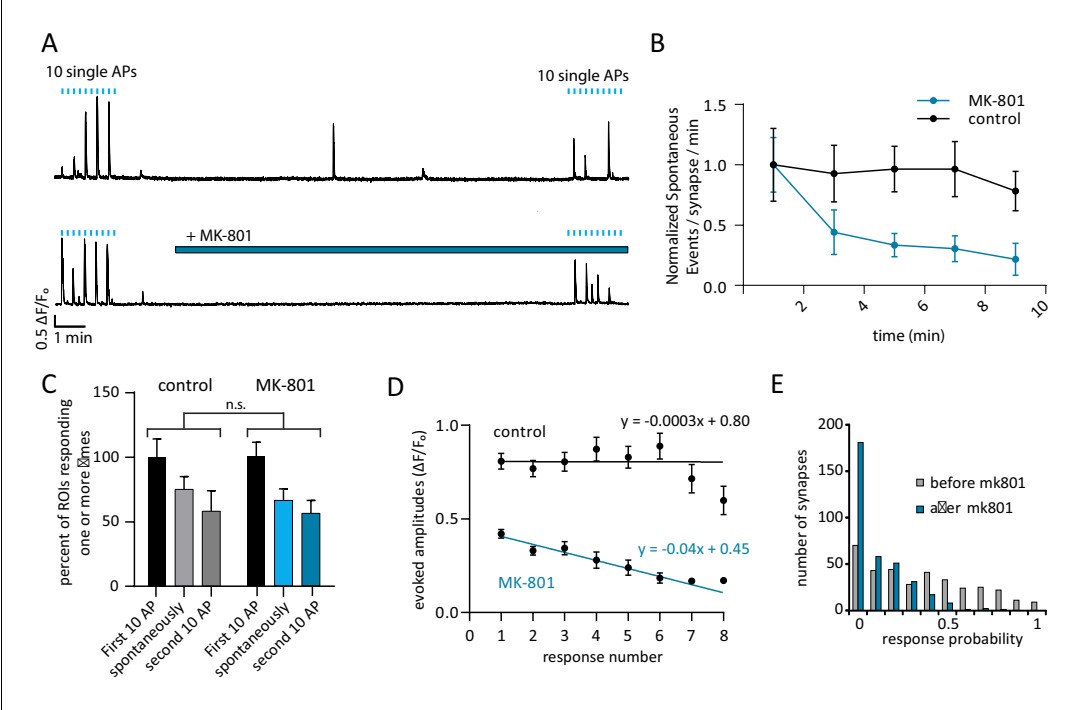

**Figure 3.** Use-dependent NMDAR blocker MK-801 can selectively silence spontaneous signals without effecting evoked responses within the same synapse. (A) Example traces outlining the experiment with (top) and without (bottom) MK-801 treatment. 10 action potentials were evoked, 10 s apart (blue tick marks). 4 min later, perfusion was changed to Tyrode's solution containing 10 μM MK-801 or vehicle. After 10 min in this solution, 10 additional action potentials are evoked, 10 s. Please note that sample traces were selected among examples that are on the larger the side of the average values (especially those after MK-801 application) for better visualization of the impact of MK-801 block on single events (B) Plot shows normalized spontaneous event frequency over the stimulation-free 10 min treatment with MK-801 or control solution. For MK-801 N = 350 ROIs from 7 cells and three cultures. For the control condition, N = 300 ROIs from 6 cells and five cultures. (C) Bars show the percentage of ROIs that respond one or more times to either the first round of 10 stimulations, the second round of 10 stimulations, or spontaneously respond during the intervening time. Control cells (vehicle treatment) and MK-801 treated cells are compared and show no significant differences. (D) Average amplitudes as $\Delta F/F_o$ for each $n^{th}$ response within an ROI (per response not per stimulation, failures to respond are skipped so 'zeros' are not averaged). For both groups, fit line generated via linear least squares method. (E) Histogram showing distribution of evoked response probabilities in cells before and after MK-801 treatment.

evoked responses. However, we find that the rate of spontaneous activity is a poor predictor of the amount of MK-801-induced decrease in evoked responses. In contrast, comparing the initial $R_p$ against the MK-801 induced change reveals that ROIs with the highest initial $R_p$ see the largest decrease in $R_p$ after treatment (*Figure 4B*). These findings are consistent with the blockade of NMDA receptors responding to stimulation during MK-801 treatment being due to evoked glutamate release alone, with little contribution from spontaneous neurotransmission. In other words, the plot shown in *Figure 4B* validates the classical assumption on the nature of MK-801 block at the single synapse level 'higher the Rp (a proxy for release probability), larger the percentage of block by MK-801'. However, this increase in block – while proportional to initial Rp – is not correlated to the amount of spontaneous events seen prior to MK-801 application (*Figure 4A*). This result indicates that a classical measure of release probability does not show a correlation between spontaneous response rate and evoked response rate at the single synapse level. Taken together, these results indicate that the majority of hippocampal synapses participate in both evoked and spontaneous neurotransmission while a small subpopulation (~20%) may only transmit spontaneous signals. The occurrence of spontaneous signals appears to be non-random which may indicate some form of presynaptic regulation (e.g. *Bal et al., 2013*). Finally, the dynamics of NMDA receptor blockade with MK-801 illustrate that in synapses that release glutamate via both modes of transmission, the postsynaptic NMDA receptors activated are largely unique to one mode or the other with limited overlap

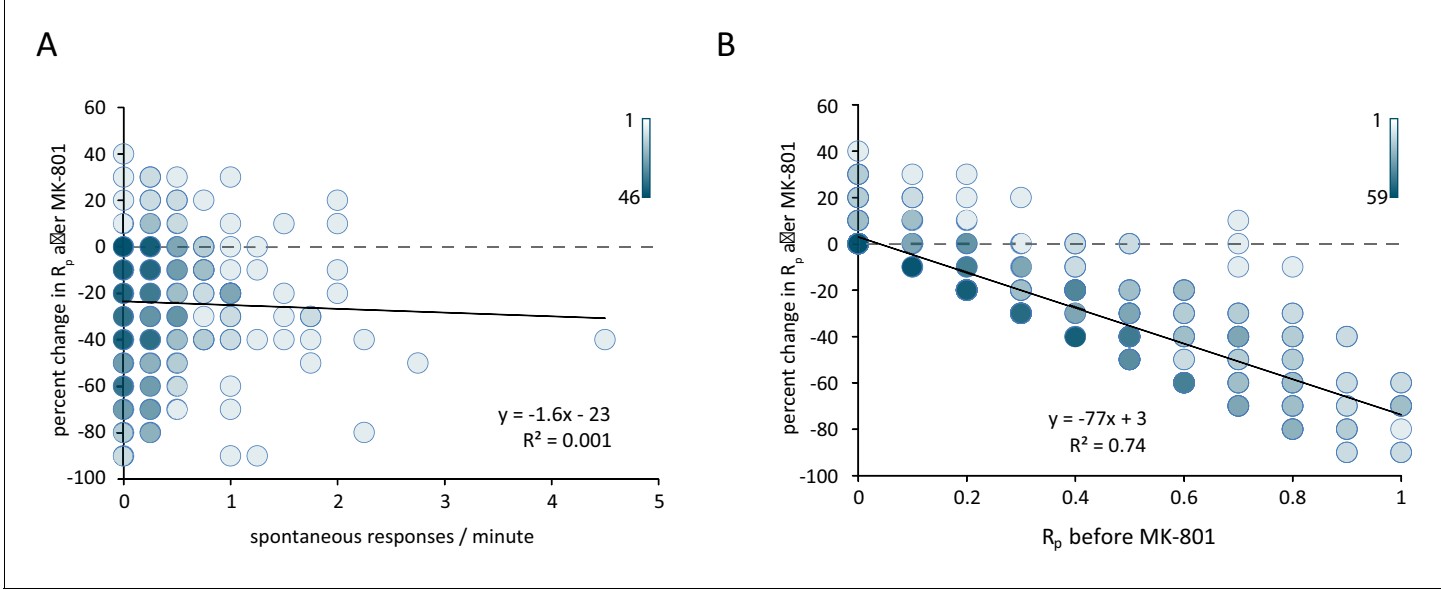

**Figure 4.** Reduction in postsynaptic responses by MK-801 correlates with evoked response probability but not spontaneous response rate. (**A**) Scatter plot of change in $R_p$ (before – after MK-801) vs spontaneous response rate before MK-801 was added. Shading indicates overlapping data points. For both plots N = 350 ROIs from 7 cells and five cultures. (**B**) Scatter plot of the change in $R_p$ after MK-801 treatment vs initial evoked response rate. Linear fit lines generated via least squares method.

within the synapse. This result implies a nanoscale segregation of sites for spontaneous versus evoked neurotransmission within individual synapses (*Atasoy et al., 2008*; *Tang et al., 2016*).

## Materials and methods

### Cell cultures
Prepared as described previously in (*Reese and Kavalali, 2015*).

### GCaMP6f-PSD95 imaging
Tyrode's solution and equipment are the same as described previously. 8 hr after transfection, 40x images were collected at five frames per second with 33% illumination intensity and a neutral density 0.6 filter in the light path. In order to maximize NMDA receptor currents, cells were imaged in $Mg^{2+}$ free solution (*Jahr and Stevens, 1990*). Stimulation was given with a 3 mm spaced parallel bipolar electrode using 30 mA of current.

### Event detection
Beginning at the first stimulation for each experiment, the image stack was divided into 30 s segments (*Figure 2*) or 10 s segments (*Figures 3* and *4*). From each segment, an average intensity projection was created and subtracted from a maximum intensity projection. This manipulation highlights puncta and eliminates background. Using this image, 2.5 µm diameter circular ROI's were placed manually over isolated puncta that did not overlap. 50 ROIs were selected in each experiment in the order that they appear, using as many segments as necessary. ROIs found to overlap multiple puncta in the following projections were relocated to measure only a single punctum. Fluorescence traces were obtained from each ROI and raw fluorescence values converted to a running $\Delta F/F_o$ using a 5 s baseline. A point was identified as a peak were identified if its $\Delta F/F_o$ was greater than 0.1, its slope calculated in 400 ms steps was greater than 687 units / sec and the peak value was greater than two standard deviations greater than the signal from the previous 2 s. Based on peak time, events that fell within a 1 s window after a stimulation were considered to be evoked and all other events were considered to be spontaneous. Relative to the presynaptic voltage deflection,

the postsynaptic $Ca^{2+}$ signal has a longer rise time and the 1 s window safely includes all evoked events. As the average spontaneous event rate is less than one event per minute, the chance of incorrectly marking a spontaneous event as an evoked event remains relatively low.

## Statistics

All statistical tests were performed using Graphpad Prism 6.01.

## Additional information

### Funding

| Funder | Author |
|---|---|
| National Institute of Neurological Disorders and Stroke | Austin L Reese<br>Ege T Kavalali |
| National Institute of Mental Health | Ege T Kavalali |

The funders had no role in study design, data collection and interpretation, or the decision to submit the work for publication.

### Author contributions

ALR, Conception and design, Acquisition of data, Analysis and interpretation of data, Drafting or revising the article; ETK, Conception and design, Analysis and interpretation of data, Drafting or revising the article

### Ethics

Animal experimentation: Animal experimentation: This study was performed in strict accordance with the recommendations in the Guide for the Care and Use of Laboratory Animals of the National Institutes of Health. All of the animals were handled according to approved institutional animal care and use committee (IACUC) protocols of the UT Southwestern Medical Center (APN# 0866-06-05-1).

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
