## [Decision Letter]

[Editors’ note: a previous version of this study was rejected after peer review, but the authors submitted for reconsideration. The first decision letter after peer review is shown below.]

Thank you for submitting your work entitled "Single synapse evaluation of the postsynaptic NMDA receptors targeted by evoked and spontaneous neurotransmission" for consideration by *eLife*. Your article has been reviewed by three peer reviewers, and the evaluation has been overseen by a Reviewing Editor and Eve Marder as the Senior Editor. The following individuals involved in review of your submission have agreed to reveal their identity: Ling-Gang Wu (Reviewer #1).

Our decision has been reached after consultation among the reviewers. Based on these discussions and the individual reviews below, we regret to inform you that your work will not be considered further for publication in *eLife*.

All the reviewers found the subject matter to be interesting and important, but all converged on a common set of concerns that not only would require substantial further experimentation to address, but which also call into question the central claim of the manuscript. Because it is unlikely that such fundamental matters can be addressed within a few weeks, as is expected for a Research Advance, we cannot consider the present manuscript further. The specific issues, which are laid out fully in the reviews, are as follows:

1) The study hinges on distinguishing spontaneous and evoked release within a single bouton, but there is insufficient evidence that the level of resolution of the techniques used is sufficient to support this claim adequately. More evidence is needed.

2) A number of quantitative factors reduce confidence in the present results, including a) apparently large differences in the mini rate compared to previous studies (from the same group), b) the properties of the indicator used, c) the low statistical power, d) possible rundown, e) analysis methods that give contradictory results on the same data, and f) analysis methods that may not represent accurate measurements of the desired quantities. All these matters would require addressing, several of which would demand additional experiments.

If you choose to undertake the extensive revisions along the lines suggested by the reviewers, we are willing to consider a new, full-length manuscript on this topic.

Reviewer #1:

In the last decade, the authors have discovered that spontaneous and evoked release are regulated separately. In the present work, the author further refines the finding by showing that at the single synapse level, spontaneous and evoked release are regulated separately by releasing to separate NMDA receptors. The finding is of significance in our understanding of synaptic strength, synaptic plasticity and homeostasis.

1) The study implies that spontaneous and evoked release are separated anatomically even within a single bouton. However, this is not supported by electron microscopic evidence. They identified single calcium transient event in postsynaptic compartment by 2.5 μm diameter ROI (in method), which is bigger than a single spine. It is hard to say "single events" in case of ROI is bigger than single spine due to records in several dendritic spines. Stronger evidence would be needed to clarify this point.

2) In Figure 2, they showed average of spontaneous response was 0.79 events/ROI/min, which is much larger than the value (~0.1 events/ROI/min) shown in a previous study from the same lab (Reese and Kavalali, 1F, GCaMP5K-PSD-95, Mg free condition). This large difference needs to be reconciled.

3) In the previous study, they used Fluo-4 which showed better results than GCaMP5K-PSD95 for detecting spontaneous calcium transient in 1.25 mM Mg, which is close to physiological condition. I wonder if the same conclusion would hold in the more physiological solution (e.g., use Fluo-4 in 1.25 mM Mg).

4) Usually, the frequency of spontaneous release is very low, less than 1 Hz, and has big variation. Therefore, it is necessary to collect enough number of examples for analysis. The cell number and culture number are in generally rather low, often in the range of 2-4. I would suggest to increase the cell number and culture number substantially to consolidate the conclusion. The authors did not show the number of experiment in Figure 3 (with large error bar). That may weaken the conclusion of Figure 3.

5) Figure 3, control, the ROIs responding to the 2nd 10AP are much less than that to the first 10AP. Does this means a significant system rundown?

Reviewer #2:

In this study, Reese and Kavalali use fluorescent calcium sensors to measure the activity of NMDA receptors in response to glutamate release. Spatially resolving these calcium signals allowed the authors to examine the likelihood that a given synaptic site 'transmits' via spontaneous and/or AP-evoked release. This is an interesting question, and it was addressed by combining the use of the activity-dependent NMDA receptor antagonist, MK-801, to illustrate any overlap between NMDA receptor activity that was driven by spontaneous versus evoked release. From these experiments the authors conclude that while many synapses participate in both evoked and spontaneous release, some synapses seem to preferentially participate in one form of release or the other.

Again, the topic of this study is of great interest, but there are a number of significant concerns that must be addressed, as detailed below.

1) Being able to clearly distinguish between NMDA-calcium signals that arise from spontaneous release versus AP driven release is paramount for the conclusions drawn in this study. As such, the authors need to justify the effectiveness of using NBQX and muscimol to preventing spontaneous APs, either through additional experiments or via very clear citations.

2) On a similar note, the use of NBQX and muscimol must also be shown to not interfere with the triggering or propagation of induced action potentials. This concern is exacerbated by the use of a bipolar electrode which is typically used to provide focused stimulation, so there is a concern that APs are not be driven throughout field of view. In order to claim that some release sites only function spontaneously and not by evoked APs, it must be demonstrated that the stimulation protocol was sufficient to always cause APs. Correlations between spontaneous and evoked rates of release may be confounded by failures to evoke APs particularly if an 'AP failure rate' is unevenly distributed in space relative to the simulating electrode.

3) The study would be much more informative if some discussion of the anatomical correlative of an ROI was provided in the main text. For example, does an ROI contain a single input or multiple inputs? And how does this parameter affect the conclusions of the study?

4) In regards to the MK-801 experiments, it was difficult to understand the various ways the data were analyzed and why certain methods were selectively applied in some cases but not in others. For example, the text for Figure 3 states that the spontaneous response rate was reduced by 87%. However, Figure 3 appears to contradict this claim, as no difference between MK801 and control conditions was apparent. Apparently, by using different methods of analysis for the same experiment, both a significant 87% reduction (vs. no change in controls) and a non-significant 25% reduction (vs 24% in controls) were observed regarding spontaneous response release in the presence of MK-801. This is particularly worrisome given a similar problem seems occur for evoked responses: Figure 3 shows no difference in ROI responders between control and MK-801 treated neurons while Figure 4 shows a nearly 50% reduction in the signal intensity for the first response after MK-801 treatment (compared to controls). This might suggest that a good portion of the individual NMDA receptors participating in evoked release, and not the entire post-synaptic NMDA pool, were indeed blocked by MK-801 during spontaneous release; these findings seem to directly contradict the primary conclusion for the figure. At the very least, a clear, persuasive argument needs to be made as to why different methods of analysis are appropriate; also, a clear, convincing explanation for their apparently different conclusions needs to be provided.

5) It is difficult to understand the main conclusions of Figure 3 through 4A and B because it is not explicitly stated how Rp was measured. If the response probability of a given ROI is measured over the entire post-treatment stimulation protocol (as in Figure 2), then stimulating in the presence of MK-801 would lower the 'Rp' causing it to decrease over the course of measuring it (and as stated, eventually lowering it to 0). Because the act of measuring significantly changes Rp, it is unclear what that number means or what it is actually measuring. The significant confound from these evoked effects (illustrated by Figure 4) makes it hard to draw or justify conclusions from comparisons with spontaneous release (as attempted in 4A). If the goal here is to examine the ability of spontaneous release to reduce 'Rp', then the analysis should only compare the first stimulation,post-treatment, or somehow otherwise remove the confounding influence that stimulation in MK-801 would have on measuring Rp.

Reviewer #3:

In this manuscript the authors report an investigation of the overlap of spontaneous and evoked transmitter release in individual synapses. The topic is controversial and its investigation therefore clearly important. The authors provide impressive imaging data showing that most synapses show both, spontaneous and stimulated events 22% of synapses show only spontaneous events. The authors claim that there is little correlation between spontaneous and evoked release, which is not so clearly explained.

While the data appears to be quite interesting, some clarification is needed:

An important question is the definition of a synapse. The Materials and methods section states that a 2.5 µm diameter ROI was placed over fluorescent puncta. How certain is it that there is only 1 synapse in such an area and that it does not include 2 synapses. This is important for the main points of the manuscript.

The authors claim that the ability to respond to evoked stimulus after MK-801..is unaffected compared to control. Figure 3 suggests an increased fraction o ROIs responding although clearly not significant. However, absence of significance is not significance of absence. On the other hand Figure 3 indicates decreased release probability. This seems contradictory and should be explained.

Related to this point: Figure 4 and the related text are not very clear.

Figure 3 legend and related text are inconsistent and confusing legend say Figure 3 top is with MK-801 but this is probably wrong. It would also help to increase the separation between top and bottom panel of Figure 3 to make clear to which trace the MK-801 bar belongs. The bar is inconsistent with the legend, which says it starts 4 min after the action potentials were evoked but it seems less than 2 min in the graph.

The term "evoked release probability" needs to be defined and probably a different term should be used because release probability is usually reserved for the probability that a vesicle of the releasable pool is released. Does it mean the probability that a synapse produces a detectable signal in response to an AP? Is that signal always from a single vesicle or could it be from multiple vesicles? The authors need to provide a clear definition.

[Editors’ note: what now follows is the decision letter after the authors submitted for further consideration.]

Thank you for submitting your article "Single synapse evaluation of the postsynaptic NMDA receptors targeted by evoked and spontaneous neurotransmission" for consideration by *eLife*. Your article has been reviewed by three peer reviewers, and the evaluation has been overseen by a Reviewing Editor and Eve Marder as the Senior Editor. The following individual involved in review of your submission has agreed to reveal his identity: Ling-Gang Wu (Reviewer #2).

The reviewers have discussed the reviews with one another and the Reviewing Editor has drafted this decision to help you prepare a revised submission.

Summary:

Previous work from this lab has provided evidence that spontaneous release of neurotransmitter and evoked release of neurotransmitter are regulated separately. The present work further shows that this separation can extend to the single synapse level. By using fluorescent calcium sensors to measure the activity of NMDA receptors with high spatial resolution, the authors tested whether a given synaptic site "transmits" via spontaneous and/or AP-evoked release. The results suggest that while many synapses participate in both evoked and spontaneous release, some synapses seem to preferentially participate in one form of release or the other, such that spontaneous and evoked release events act at separate NMDA receptors. The finding is of significance to the understanding of synaptic strength, synaptic plasticity and homeostasis.

Essential revisions:

While there was considerable enthusiasm for the manuscript and the response to previous comments, the three reviewers were largely in agreement about issues that still require addressing. Two major points emerged. 1) All reviewers found that the rebuttal letter was well written and convincing, but all agreed that the manuscript itself had not been sufficiently edited to ensure that the important information provided in the response to reviewers would be available to readers of the manuscript. Please incorporate the figures and information from the rebuttal either directly into the manuscript main text or add them as supplementary figures. 2) Specific concerns remained about Figure 3 and Figure 4 regarding self-consistency and interpretability.

The combined reviews with key points from the consultation session are included below to facilitate your revision.

Combined reviewers' points regarding essential revision 1:

– The rebuttal letter is, overall, very good. But, the text of the manuscript does not appear to have been changed in any significant way to reflect the issues that were raised, and largely addressed, in the reviews and response to the reviews. Specifically, all three referees asked for detailed discussion of the "ROI" – a key question was: are these single synapses or do they contain multiple synapses? This issue is addressed in the rebuttal but not in the manuscript. Our concerns about the stimulation protocols and recording conditions were also addressed in the rebuttal but not in the revised manuscript. This is an interesting study, but the issues raised do need to be addressed in the revised text of the manuscript.

– The manuscript is now much clearer but the some of the figures in the rebuttal with detailed description should be part of the manuscript. Some could be supplementary information. Figure 8, lower figure may need detailed description and analysis to make the point. What is the vertical scale in these graphs?

Combined reviewers' points regarding essential revision 1:

– In addition, Figure 3 and Figure 4 still remain confusing – we asked for clarification but clarification was not provided: we still do not know what the bars, labeled "spontaneous" in Figure 3, or how they relate to the graph show in in Figure 3.

– The clarification of Figure 3 and Figure 4 is essential. While Figure 3 suggests no effect of MK-801 on spontaneous responses, Figure 3 does. I can somehow imagine what this means but it needs to be spelled out clearly in the text.

– The ∆F/F0 values in Figure 3 seem almost an order of magnitude smaller than in the example traces of panel A. Is the scale bar in A wrong or the values in D or do the traces show unrepresentative signal amplitudes?

– I still do not understand the arguments surrounding Figure 4. If MK-801 were to block all evoked responses entirely, all data points would lie on the line from x,y = 0,0 to 1,-1 simply because the maximum -∆Rp equals Rp before. The figure shows a rather linear slope of -0.8 suggesting 80% inhibition of Rp whatever it was before MK-801 was added and I do not recognize any relation between blocking efficiency with Rp. ∆Rp may therefore also not be a good choice as analyzed parameter in panel A. Maybe the relative change ∆Rp/Rp(before) might be more instructive.

[Editors' note: further revisions were requested prior to acceptance, as described below.]

Thank you for resubmitting your work entitled "Single synapse evaluation of the postsynaptic NMDA receptors targeted by evoked and spontaneous neurotransmission" for further consideration at *eLife*. Your revised article has been favorably evaluated by Eve Marder (Senior editor), a Reviewing editor, and three reviewers.

The manuscript is greatly improved but there are some minor points that must be addressed before acceptance, as outlined below:

There remains some confusion about Figure 4. One reviewer points out that the fact that the data fall on a line suggest that MK801 blocks about 80% of the channels regardless of initial Rp; the percent block necessarily decreases as Rp decreases because there are fewer channels activated that can be blocked. Another reviewer suggests that this is actually the point: MK801 blocking efficiency may be constant at ~80% but the fact that it is predictable from evoked Rp but not spontaneous rate supports the separation of the groups. Since the reviewers had such difficulty working this out, a clarification in the text would be helpful. Also, Figure 4 is not called in the text. Please add a reference to this plot at the appropriate place in the manuscript. Finally, a minor question remains regarding the relationship between the amplitudes of the example signal in Figure 3 and the mean summary data in Figure 3 (see reviewer comment below). Please indicate whether the traces shown unusually large or whether there is an error in the scale bar.

Additional comments from the reviewers are included below so that you can see the nature of the confusion of the reviewers.

The response to the following point is not clear to me:

"I still do not understand the arguments surrounding Figure 4. If MK-801 were to block all evoked responses entirely, all data points would lie on the line from x,y = 0,0 to 1,-1 simply because the maximum -ΔRp equals Rp before. The figure shows a rather linear slope of -0.8 suggesting 80% inhibition of Rp whatever it was before MK-801 was added and I do not recognize any relation between blocking efficiency with Rp. ΔRp may therefore also not be a good choice as analyzed parameter in panel A. Maybe the relative change ΔRp/Rp(before) might be more instructive."

While the vertical scale of Figure 4 was changed by the authors, the data points are still the same and give in my opinion a misleading impression. The graph does not show ∆Rp/Rp(before). Such a graph will show points scattered around a constant value of ~0.8 for all Rp before values, indicating that the blocking efficiency is independent of Rp. I do not understand what Figure 4 tells us.

The review stated:

"The ΔF/F0 values in Figure 3 seem almost an order of magnitude smaller than in the example traces of panel A. Is the scale bar in A wrong or the values in D or do the traces show unrepresentative signal amplitudes?"

In the rebuttal the authors write:

"The example trace in Figure 1 was chosen for aesthetic reasons and is not a good representative of the average amplitudes. A new representative trace has been chosen that closely matches the average amplitude of 0.67 dF/F for that experiment."

However, the discrepancy the review pointed out is not between Figure 3 and Figure 1 but between panels A and D of Figure 3 because panel A presumably shows representative recorded data that led to the quantification shown in in panel D.

---

## [Author Response]

[Editors’ note: the author responses to the first round of peer review follow.]

*All the reviewers found the subject matter to be interesting and important, but all converged on a common set of concerns that not only would require substantial further experimentation to address, but which also call into question the central claim of the manuscript. Because it is unlikely that such fundamental matters can be addressed within a few weeks, as is expected for a Research Advance, we cannot consider the present manuscript further. The specific issues, which are laid out fully in the reviews, are as follows:*

*1) The study hinges on distinguishing spontaneous and evoked release within a single bouton, but there is insufficient evidence that the level of resolution of the techniques used is sufficient to support this claim adequately. More evidence is needed.*

We now clarified our criteria for selection of putative single synapses (regions of interest or ROIs). In our response, we also included additional measures supporting our assumption to a large extent we are recording from single boutons.

*2) A number of quantitative factors reduce confidence in the present results, including a) apparently large differences in the mini rate compared to previous studies (from the same group), b) the properties of the indicator used, c) the low statistical power, d) possible rundown, e) analysis methods that give contradictory results on the same data, and f) analysis methods that may not represent accurate measurements of the desired quantities. All these matters would require addressing, several of which would demand additional experiments.*

We tried to address all these issues in the revised manuscript. Please see our specific responses to the reviewers’ questions.

*Reviewer #1:*

*In the last decade, the authors have discovered that spontaneous and evoked release are regulated separately. In the present work, the author further refines the finding by showing that at the single synapse level, spontaneous and evoked release are regulated separately by releasing to separate NMDA receptors. The finding is of significance in our understanding of synaptic strength, synaptic plasticity and homeostasis.*

*1) The study implies that spontaneous and evoked release are separated anatomically even within a single bouton. However, this is not supported by electron microscopic evidence. They identified single calcium transient event in postsynaptic compartment by 2.5 μm diameter ROI (in method), which is bigger than a single spine. It is hard to say "single events" in case of ROI is bigger than single spine due to records in several dendritic spines. Stronger evidence would be needed to clarify this point.*

Due to the short format of the earlier submission, we allocated very limited space to method details surrounding data processing. We would like to take this opportunity to elaborate on the data collection process for this study. The following paragraph is now added to the Materials and methods section:

“Beginning at the first stimulation for each experiment, the image stack was divided into 30 second segments (Figure 2) or 10 second segments (Figure 3, Figure 4). From each segment, an average intensity projection was created and subtracted from a maximum intensity projection. This manipulation highlights puncta and eliminates background. Using this image, 2.5 µm diameter circular ROI’s were placed manually over isolated puncta that did not overlap. 50 ROIs were selected in each experiment in the order that they appear, using as many segments as necessary. ROIs found to overlap multiple puncta in the following projections were relocated to measure only a single punctum.”

To better demonstrate this process, we now include two example images from an experiment in Figure 1. Rather than projecting a 30 second time window, a projection was made from the entire experiment to highlight all puncta. The left image is the projection alone, and the right image includes the ROI selections.

We considered the possibility that multiple synapses may appear as one puncta. If this were to be true, we would expect two specific behaviors. First, if two synapses were to appear in the same ROI, it is possible that effected ROIs would have a higher response probability, a higher spontaneous response rate or roughly double amplitude when responding to evoked stimuli together. As seen in Figure 1, there is little correlation between response probability and spontaneous response rate. We would expect that ROIs containing two or more synapses would have considerably more of both evoked and spontaneous responses and would thus show as a trend on this plot. This is not the case. Second, if we assume multiple synapses per ROI, we would also expect that the relative strength differences of the included synapses would produce significantly higher amplitude variance at the effected ROI. To look for a population of synapses where this is true, we compared both the spontaneous response rate and evoked response probability against the coefficient of variation (Cv) for amplitudes measured from all synapses containing 2 or more responses in Figure 5. 294 ROIs had 2 or more spontaneous events and 151 ROIs had 2 or more evoked events.

Author response image 1.**DOI:**
http://dx.doi.org/10.7554/eLife.21170.008

In the plots, we see very little correlation between the coefficient of amplitude variation and the frequency of evoked or spontaneous events. We do not see evidence for a high frequency, high amplitude variance population of ROIs. We think this analysis supports the premise that a majority of the ROI selected for analysis contain a single synapse.

*2) In Figure 2, they showed average of spontaneous response was 0.79 events/ROI/min, which is much larger than the value (~0.1 events/ROI/min) shown in a previous study from the same lab (Reese and Kavalali, 1F, GCaMP5K-PSD-95, Mg free condition). This large difference needs to be reconciled.*

This mismatch is best explained by differences in GCaMP constructs. Compared to GCaMP5k, the newer GCaMP6f has a higher affinity for free Ca^2+^ ions (Kd = 114 nM for GCaMP6f vs. 189 nM for GCaMP5K) and higher dynamic range (max in vivo ∆F/Fₒ = 9.4 ± 0.14 for GCaMP5k and 19 ± 2.8 for GCaMP6f) (Akerboom, Chen et al. 2012, Chen, Wardill et al. 2013). Using GCaMP5K, we had detected relatively fewer peaks most likely due to the fact that many peak amplitudes barely surpassed our detection threshold. Comparatively, GCaMP6f produced very robust, high amplitude signals that readily surpassed the detection threshold.

Author response image 2.**DOI:**
http://dx.doi.org/10.7554/eLife.21170.009

*3) In the previous study, they used Fluo-4 which showed better results than GCaMP5K-PSD95 for detecting spontaneous calcium transient in 1.25 mM Mg, which is close to physiological condition. I wonder if the same conclusion would hold in the more physiological solution (e.g., use Fluo-4 in 1.25 mM Mg).*

In our previous publication, we chose Fluo-4 AM specifically for its ability to detect mSCTs in 1.25 mM Mg^2+^. However, Fluo-4 AM was unsuitable to detect NMDA currents alone in the presence of Mg^2+^ as we tested in the previous Figure 4. Without the contribution of ER stores to the calcium transients, less than 5% of the total number of events were detectable compared to events in Mg^2+^ free solution. This result suggests that while Fluo-4 AM can detect NMDA transients without ER store contribution, it is only feasible in the absence of Mg^2+^. Under ER store block and Mg^2+^ free conditions, Fluo-4 AM detects transients with amplitudes of 0.07 ± 0.009 ∆F/Fₒ which is significantly smaller than GCaMP6F. We believe the lower amplitudes represents a lower detection fidelity especially given that it produces low spontaneous release estimates compared to GCaMP6f (0.20 ± 0.06 events/ROI/min with Fluo-4 AM vs 0.79 ± 0.04 events/ROI/min with GCaMP6f). Additionally, for the purposes of the current study, we required that only a single cell be labeled so as not to conflate presynaptic and postsynaptic Ca^2+^ signals during stimulation. For this experimental setup, Fluo-4 would need to be loaded into a cell via patch pipette which causes significant dialysis of the cytosol, especially over the longer recording periods required for Figure 3. For these reasons GCaMP6f was chosen as the best indicator for this study.

*4) Usually, the frequency of spontaneous release is very low, less than 1 Hz, and has big variation. Therefore, it is necessary to collect enough number of examples for analysis. The cell number and culture number are in generally rather low, often in the range of 2-4. I would suggest to increase the cell number and culture number substantially to consolidate the conclusion. The authors did not show the number of experiment in Figure 3 (with large error bar). That may weaken the conclusion of Figure 3.*

We have added additional experiments to Figure 2, Figure 3 and Figure 4 in order to bolster the statistical power of our measurements. Figure 2 now has 3 additional experiments, Figure 3 has 2 additional treatment and 2 additional control experiments. Figure 4 includes 2 additional experiments. The number of ROIs, cells and cultures for each experiment are reported in each figure legend.

*5) Figure 3, control, the ROIs responding to the 2nd 10AP are much less than that to the first 10AP. Does this means a significant system rundown?*

Though the illumination during recordings is minimal, photobleaching and rundown do occur. For this reason the control condition in Figure 3 was necessary to separate the effects of MK-801 from photobleaching and rundown.

*Reviewer #2:*

In this study, Reese and Kavalali use fluorescent calcium sensors to measure the activity of NMDA receptors in response to glutamate release. Spatially resolving these calcium signals allowed the authors to examine the likelihood that a given synaptic site 'transmits' via spontaneous and/or AP-evoked release. This is an interesting question, and it was addressed by combining the use of the activity-dependent NMDA receptor antagonist, MK-801, to illustrate any overlap between NMDA receptor activity that was driven by spontaneous versus evoked release. From these experiments the authors conclude that while many synapses participate in both evoked and spontaneous release, some synapses seem to preferentially participate in one form of release or the other.

*Again, the topic of this study is of great interest, but there are a number of significant concerns that must be addressed, as detailed below.*

*1) Being able to clearly distinguish between NMDA-calcium signals that arise from spontaneous release versus AP driven release is paramount for the conclusions drawn in this study. As such, the authors need to justify the effectiveness of using NBQX and muscimol to preventing spontaneous APs, either through additional experiments or via very clear citations.*

GABAA receptors are highly expressed in multiple pyramidal cell types of the hippocampus where they are localized to the soma and dendrites but not axons (Sperk, Schwarzer et al. 1997). In high density dissociated cultures we have found muscimol and NBQX to be excellent inhibitors of network activity. To better demonstrate this point, we obtained current clamp recordings where cells were recorded in the same Tyrode’s solution as the imaging experiments before perfusing in solution containing muscimol and NBQX. Within seconds, the combination of muscimol and NBQX halt action potential firing (arrows) and produce a consistent baseline. Under these conditions, excitatory input is blocked or shunted well enough to damp out activity at the soma.

Additionally, data from Figure 2 can be used to illustrate the reliability of this method. Four example experiments are plotted in Figure 7. Each mark represents a detected event and the red ticks above indicate where stimulation was given. Obvious event clustering is seen every 30 seconds when the stimulation is given without any reverberatory action potentials or other obvious network activity. Despite the inhibition of network activity, stimulation can reliably trigger action potential driven events.

Author response image 3.**DOI:**
http://dx.doi.org/10.7554/eLife.21170.010

*2) On a similar note, the use of NBQX and muscimol must also be shown to not interfere with the triggering or propagation of induced action potentials. This concern is exacerbated by the use of a bipolar electrode which is typically used to provide focused stimulation, so there is a concern that APs are not be driven throughout field of view. In order to claim that some release sites only function spontaneously and not by evoked APs, it must be demonstrated that the stimulation protocol was sufficient to always cause APs. Correlations between spontaneous and evoked rates of release may be confounded by failures to evoke APs particularly if an 'AP failure rate' is unevenly distributed in space relative to the simulating electrode.*

Reliable stimulation is indeed a critical assumption throughout our study. In order to ensure reliable AP generation we utilize a parallel bipolar electrode rather than the concentric bipolar electrode commonly used to provide focal stimulation. The 3 mm gap between the poles of this electrode is wide enough to encompass the entire recording field and readily stimulates action potentials across the majority of the coverslip.

To illustrate this, in Figure 8 are two example imaging experiments using Syb2-GCaMP6f. This GCaMP-synaptobrevin fusion is trafficked to presynaptic compartments and provides Ca^2+^ signals due to action potential driven Ca^2+^ influx. These recordings are made in the presence of CNQX and AP5 to prevent reverberatory network spiking upon stimulation. These images were collected on the same rig as the submitted manuscript, and utilize the same 0.1 Hz stimulation protocol

Author response image 4.**DOI:**
http://dx.doi.org/10.7554/eLife.21170.011

In the example experiments, the location of each ROI in the visual field is displayed in the left panel. The response probability is displayed for each ROI in both experiments as a function of both X and Y axis on the right (error bars are standard deviation). A one-way ANOVA of this data comparing ROIs grouped by quadrant as displayed in the left panels shows no significant difference in stimulation efficacy across the visual field. The amplitudes from a single ROI using this technique are typically less than 0.05 ∆F/Fₒ, so we expect that the distribution of response probabilities towards the upper limit represents a detection failure rather than an action potential failure. Additionally, because the Rp values reported in Figure 2 and Figure 3 do show individual ROIs approaching and equal to 1, we expect that our stimulation protocol is not only reliable throughout the visual field but has an acceptably low AP failure rate.

*3) The study would be much more informative if some discussion of the anatomical correlative of an ROI was provided in the main text. For example, does an ROI contain a single input or multiple inputs? And how does this parameter affect the conclusions of the study?*

To better clarify the intentions of our method, the following detail has been added to the first paragraph of the Results section:

“The indicator was expressed in the postsynaptic compartment where its clustering produced punctate signals consistent with a postsynaptic localization (Figure 1). Fluorescence traces were generated from ROIs (regions of interest) placed over individual puncta so signal peaks corresponding to synaptic activity could be counted. “

As discussed in depth in our response to reviewer #1, we believe that evidence supports our assertion that in this system a single ROI corresponds to a single synapse. However, in an effort to maintain minimal assumptions we choose to refer to our data points as ROIs rather than synapses. Likewise we have chosen to express each ROI’s evoked responses using term Rp instead of the presynaptic measurement Pr.

*4) In regards to the MK-801 experiments, it was difficult to understand the various ways the data were analyzed and why certain methods were selectively applied in some cases but not in others. For example, the text for Figure 3 states that the spontaneous response rate was reduced by 87%. However, Figure 3 appears to contradict this claim, as no difference between MK801 and control conditions was apparent. Apparently, by using different methods of analysis for the same experiment, both a significant 87% reduction (vs. no change in controls) and a non-significant 25% reduction (vs 24% in controls) were observed regarding spontaneous response release in the presence of MK-801.*

In the case of Figure 3, these comparisons are posed to illustrate the differential effects of MK-801 upon spontaneous and evoked responses in this experimental protocol. Subsection “Use-dependent NMDAR block reveals sub-synaptic separation of evoked and spontaneous signals” is in reference to the ability of MK-801 to block spontaneous responses as shown in Figure 3. The spontaneous response rate per minute was decreased 87% by MK-801 treatment when comparing the first minute with the last minute of the 10 minute period where no stimulation was given. When comparing the untreated cells, there was no significant difference in the frequencies when we compared the first and last minute. This comparison illustrates that when treated with MK-801, the population of synapses that respond spontaneously are in fact experiencing a decrease in response frequencies consistent with spontaneously released glutamate being unable to activate NMDARs that were blocked by MK-801 during a previous spontaneous response. The sentence in question has been edited for clarity and now reads:

“During the 10 minutes of stimulation-free MK-801 treatment, spontaneous response rate decreased by 87% which is consistent with MK-801 induced block of NMDARs that respond spontaneously (Figure 3) (p = 0.009 comparing first and last minute of MK-801 with student’s T-test).”

The data displayed in Figure 3 is a comparison of the number of ROIs that respond one or more times during the second train of action potentials (labeled second 10 AP) or in the intervening period where no stimulation was given (labeled spontaneous) when only considering ROIs that respond to the first stimulation. In this way, ROI’s that exhibited spontaneous but not evoked responses were not considered as they are irrelevant to this analysis. The purpose of this figure is to illustrate that even though MK-801 is producing significant decreases in spontaneous response frequency as seen in Figure 3, this does not occlude the ability of these synapses to respond to a second stimulation after treatment.

This comparison is made this way to ask if blocking spontaneous responses with MK-801 would prevent further evoked responses entirely. It is important to note that MK-801 was not washed out of the bath solution before the second train of action potentials (as illustrated in Figure 3). Keeping MK-801 in the bath is meant to prevent the confounding situation that would arise if MK-801 started to wash out during the second train of stimulations, which could lead to a reversal of its blockade and an underestimation of NMDAR population overlap. Since MK-801 did remain in the bath, it was important to count synapses that responded one or more times because the amplitude of each subsequent response decreases as illustrated in Figure 3. In order to control for run-down and photobleaching during this experiment, MK-801 treatment is compared to the control condition rather than directly comparing the first and second stimulation trains within each condition.

To better convey this fact, the Y axis label for Figure 3 is now changed to read “percent of ROIs responding one or more times” and references to more specific portions of the figure have been added to the text.

*This is particularly worrisome given a similar problem seems occur for evoked responses: Figure 3 shows no difference in ROI responders between control and MK-801 treated neurons while Figure 4 shows a nearly 50% reduction in the signal intensity for the first response after MK-801 treatment (compared to controls). This might suggest that a good portion of the individual NMDA receptors participating in evoked release, and not the entire post-synaptic NMDA pool, were indeed blocked by MK-801 during spontaneous release; these findings seem to directly contradict the primary conclusion for the figure. At the very least, a clear, persuasive argument needs to be made as to why different methods of analysis are appropriate; also, a clear, convincing explanation for their apparently different conclusions needs to be provided.*

As is true for spontaneously active ROIs, the comparison between ROIs that respond one or more times during stimulation between MK-801 treated cells and control in Figure 3 illustrates that the number of synapses able to respond is not different. Similar to the situation described above regarding spontaneous responses, the fact that an ROI can respond after MK-801 treatment does not mean that its response amplitudes are unchanged.

MK-801 does in all cases have an effect on event amplitudes due to the nature of its action. MK-801 blocks the NMDA receptor by binding to a site that is only accessible when the channel is open thereby making it a use-dependent blocker (Huettner and Bean 1988). As such, the presence of MK-801 shortens NMDAR mean open times, which results in a decrease in peak fluorescence amplitudes (Atasoy, Ertunc et al. 2008). Ca^2+^ ions diffusion rate across the spine neck is limited which causes the fluorescence signal peak to more closely correlate with the integral of the NMDA EPSC rather than the EPSC amplitude (Sabatini, Oertner et al. 2002, Bloodgood and Sabatini 2005). We believe this effect to be the reason that Figure 3 displays decreased first response amplitudes in the presence of MK-801. However, Figure 3 is only meant to illustrate that average event amplitudes decrease linearly with each successive response. While it is tempting to compare amplitudes before/after as a proxy for the number of NMDARs blocked, we chose not to directly compare event amplitudes for the purposes of this manuscript. This avoids conflating the MK-801 driven decrease in amplitude with photobleaching which becomes increasingly non-linear due to its uneven effects on fluorescent versus quenched fluorophores.

*5) It is difficult to understand the main conclusions of Figure 3 through 4A and B because it is not explicitly stated how Rp was measured. If the response probability of a given ROI is measured over the entire post-treatment stimulation protocol (as in Figure 2), then stimulating in the presence of MK-801 would lower the 'Rp' causing it to decrease over the course of measuring it (and as stated, eventually lowering it to 0). Because the act of measuring significantly changes Rp, it is unclear what that number means or what it is actually measuring. The significant confound from these evoked effects (illustrated by Figure 4) makes it hard to draw or justify conclusions from comparisons with spontaneous release (as attempted in 4A). If the goal here is to examine the ability of spontaneous release to reduce 'Rp', then the analysis should only compare the first stimulation,post-treatment, or somehow otherwise remove the confounding influence that stimulation in MK-801 would have on measuring Rp.*

In the experiments shown in Figure 2, the response probability is measured over the entire experiment as there is no drug treatment. In this case the response probability, Rp, is the number of responses divided by the number of stimulations (15). For our purposes, Rp is quite literally the probability of response at a given ROI. In order to observe changes in response probability due to the effects of MK-801, we have made this measurement both before and after MK-801 treatment for the experiments in Figure 3 and Figure 4 by utilizing two separate trains of 10 action potentials. Rp measurements do not span across the MK-801 treatment.

The concern over MK-801 directly effecting the response probability by altering response amplitudes is the reason why we have chosen to display the data from this experiment as the comparison in Figure 3 as well as the comparisons in Figure 3 / 4A,B. Figure 3 seeks to discount the effect of MK-801 on calcium transient amplitudes by displaying the proportion of synapses that respond in each category. However, this does not take into account the fact that MK-801 affects the second Rp measurement. In order to accurately measure the effects of MK-801 blockade during the 10 minute spontaneous only treatment, MK-801 was kept in the bath solution during the second Rp measurement. Had this not been the case, drug wash-out and NMDAR recovery would have underestimated the magnitude of MK-801 block.

Performing the experiment with MK-801 present during the second stimulation does still allow us to make the comparisons in Figure 4. With MK-801 in the bath solution, we may assume that the open channel block affects all NMDARs equally. With the assumption that the rate of NMDAR block is the same per each NMDA opening, we can ask if the amount of MK-801 blockade (here, the change in Rp between the first and second measurements or ∆Rp) is strongly correlated with each ROIs spontaneous response rate (Figure 4) or its evoked response probability (Figure 4). Our finding that the amount of block does not correlate with the rate of spontaneous responses suggests that this is not a driving factor in our observed changes. Instead the conclusion of this experiment is that the amount of MK-801 block of evoked responses seems to only depend on the amount of evoked responses and thus the amount of opportunities that MK-801 has to block NMDARs responding to evoked release alone. Similarly, the number of chances that MK-801 has to block NMDARs responding to spontaneous release does not appear to affect evoked responses. We believe that this is functional evidence for separate populations of NMDARs being activated by evoked and spontaneous glutamate release in close spatial proximity.

*Reviewer #3:*

*In this manuscript the authors report an investigation of the overlap of spontaneous and evoked transmitter release in individual synapses. The topic is controversial and its investigation therefore clearly important. The authors provide impressive imaging data showing that most synapses show both, spontaneous and stimulated events 22% of synapses show only spontaneous events. The authors claim that there is little correlation between spontaneous and evoked release, which is not so clearly explained.*

*While the data appears to be quite interesting, some clarification is needed:*

*An important question is the definition of a synapse. The Materials and methods section states that a 2.5 µm diameter ROI was placed over fluorescent puncta. How certain is it that there is only 1 synapse in such an area and that it does not include 2 synapses. This is important for the main points of the manuscript.*

The interpretation of our measurements does indeed depend on our ability to choose one synapse per ROI. Although, the interpretation of our MK-801 block experiments is not as sensitive to the validity of this assumption. Our analysis suggests that our methods enable us to select ROIs containing putative single synapses. In our response to reviewer #1 above, we outlined several reasons why we believe this to be the case

*The authors claim that the ability to respond to evoked stimulus after MK-801..is unaffected compared to control. Figure 3 suggests an increased fraction o ROIs responding although clearly not significant. However, absence of significance is not significance of absence. On the other hand Figure 3 indicates decreased release probability. This seems contradictory and should be explained.*

Data from 4 additional experiments has been added to this dataset totaling 200 additional ROIs. As reported, a one-way ANOVA does not detect any differences between the control and MK-801 treated conditions. Additionally, we have run a two-way ANOVA with Sidak’s multiple comparisons to look for differences between groups per each measurement and found no differences.

Figure 3 is a comparison of the Rp in the MK-801 treatment group measured before and after the drug was perfused. We have included a more detailed discussion of this topic in response to the comments by reviewer #2. Briefly, MK-801 washout and subsequent recovery from blockade would lead to an underestimation of the number of NMDARs blocked by spontaneous responses in the presence of the drug. In order to avoid this confound, we allowed MK-801 to remain in the bath during the second Rp measurement. In this experiment, MK-801 does block NMDARs as they open and affects the second Rp measurement by this mechanism. This experimental procedure still allows us to compare the rate of block during the second stimulus train between ROIs that have high or low spontaneous response rates and to draw conclusions from these correlations.

*Related to this point: Figure 4 and the related text are not very clear.*

We now re-wrote the text for clarity.

*Figure 3 legend and related text are inconsistent and confusing legend say Figure 3 top is with MK-801 but this is probably wrong. It would also help to increase the separation between top and bottom panel of Figure 3 to make clear to which trace the MK-801 bar belongs. The bar is inconsistent with the legend, which says it starts 4 min after the action potentials were evoked but it seems less than 2 min in the graph.*

Due to a graphing error, the scale bar was in fact too long. We apologize for this error and have corrected it in the revised manuscript. The example traces in Figure 3 have been edited to make it more apparent which trace represents the control.

*The term "evoked release probability" needs to be defined and probably a different term should be used because release probability is usually reserved for the probability that a vesicle of the releasable pool is released. Does it mean the probability that a synapse produces a detectable signal in response to an AP? Is that signal always from a single vesicle or could it be from multiple vesicles? The authors need to provide a clear definition.*

For the purposes of this manuscript we have chosen to distinguish between the fusion of presynaptic vesicles commonly referred to using the term “release” and the signals that we detect which we term “responses”. For this same reason, we do not report the propensity of a ROI respond to stimulation as release probability but rather response probability. Since these are postsynaptic measurements, we are not actively monitoring synaptic release but rather the resulting postsynaptic responses. We prefer not to conflate the two measurements. In the text, when referring to our measurements we use the term “response” and when referring to the physiological process of vesicle fusion we use the term “release”.

Having checked the manuscript, we found one usage error and have corrected it. To better differentiate between these concepts we have added the following to the sentence:

“To assess if spontaneous response rate and evoked response probability (reported here as Rp due to the postsynaptic nature of our measurement and so as not to be confused with presynaptic Pr) may be functionally correlated, cells were recorded for 8 minutes during which single action potentials were stimulated every 30 seconds for a total of 15 action potentials (see Video 1).”

[Editors' note: the author responses to the re-review follow.]

*Essential revisions:*

*While there was considerable enthusiasm for the manuscript and the response to previous comments, the three reviewers were largely in agreement about issues that still require addressing. Two major points emerged. 1) All reviewers found that the rebuttal letter was well written and convincing, but all agreed that the manuscript itself had not been sufficiently edited to ensure that the important information provided in the response to reviewers would be available to readers of the manuscript. Please incorporate the figures and information from the rebuttal either directly into the manuscript main text or add them as supplementary figures. 2) Specific concerns remained about Figure 3 and Figure 4 regarding self-consistency and interpretability.*

*The combined reviews with key points from the consultation session are included below to facilitate your revision.*

*Combined reviewers' points regarding essential revision 1:*

*– The rebuttal letter is, overall, very good. But, the text of the manuscript does not appear to have been changed in any significant way to reflect the issues that were raised, and largely addressed, in the reviews and response to the reviews. Specifically, all three referees asked for detailed discussion of the "ROI" – a key question was: are these single synapses or do they contain multiple synapses? This issue is addressed in the rebuttal but not in the manuscript. Our concerns about the stimulation protocols and recording conditions were also addressed in the rebuttal but not in the revised manuscript. This is an interesting study, but the issues raised do need to be addressed in the revised text of the manuscript.*

In order to address the reviewers’ concerns within the text of the manuscript, we add a supplement to Figure 2. Since the data displayed in this figure primarily addresses the methods used, we have included these panels as supplemental information rather than part of the main figures.

*– The manuscript is now much clearer but the some of the figures in the rebuttal with detailed description should be part of the manuscript. Some could be supplementary information. Figure 8, lower figure may need detailed description and analysis to make the point. What is the vertical scale in these graphs?*

We have added the scatter plot from the previous response letter page 6 as example data in Figure 2, where it illustrates the layout of the experiments. The data for this figure is plotted exactly on its x value (time) but is scattered randomly within its row along the y axis to facilitate visualization of multiple overlapping responses. The other plots have been combined in the supplement to Figure 2, where they provide additional information about the method used. We have also included dye-based Ca^2+^ imaging experiment that illustrates that NBQX and muscimol prevent reverberatory network excitability without preventing synaptic responses. We believe this data in conjunction with the fact that as illustrated in the new Figure 2, we observe reliable and synchronous synaptic responses upon stimulation show that our protocols do not hinder stimulation or generate uncontrolled spiking. The presynaptic Ca^2+^ imaging is not included in the revised manuscript because it is not yet a published technique for the lab. With the reviewer’s permission, we would prefer to leave this data in the earlier response letter.

*Combined reviewers' points regarding essential revision 1:*

*– In addition, Figure 3 and Figure 4 still remain confusing – we asked for clarification but clarification was not provided: we still do not know what the bars, labeled "spontaneous" in Figure 3, or how they relate to the graph show in in Figure 3.*

In order to clarify this issue in Figure 3, we have changed the y-axis in Figure 3 to read “percent of ROIs that respond one or more times” from the original wording. We hope that this conveys that these counts refer the number of ROIs with any non-zero number of responses rather than an event rate per ROI as displayed in Figure 3. Likewise, we have changed the axis in Figure 3 to read “evoked amplitudes” to clarify that those numbers are from stimulated responses rather than spontaneous ones.

In the text, the figure legends for both 3B,C and D have been edited for clarity and now read:

“B. Plot shows normalized spontaneous event frequency over the 10 minute treatment with MK-801 or control solution (not during stimulation trains). For MK-801 N = 350 ROIs from 7 cells and 3 cultures. For the control condition, N = 300 ROIs from 6 cells and 5 cultures. C. Bars show the percentage of ROIs that respond one or more times to either the first round of 10 stimulations, the second round of 10 stimulations, or spontaneously respond during the intervening time. Control cells (vehicle treatment) and MK-801 treated cells are compared and show no significant differences. D. Average amplitudes as ∆F/Fₒ for each n^th^ response within an ROI (per response not per stimulation, failures to respond are skipped and “zeros” are not averaged).”

*– The clarification of Figure 3 and Figure 4 is essential. While Figure 3 suggests no effect of MK-801 on spontaneous responses, Figure 3 does. I can somehow imagine what this means but it needs to be spelled out clearly in the text.*

Figure 3 reports the number of ROIs while Figure 3 reports the rate of events detected at those ROIs. To clarify this point, the following sentence has been added to the Results section:

“The application of MK-801 did not affect the number of spontaneously active ROIs (Figure 3) though it did significantly decrease the rate of detected events at those ROIs (Figure 3).”

*– The ∆F/F0 values in Figure 3 seem almost an order of magnitude smaller than in the example traces of panel A. Is the scale bar in A wrong or the values in D or do the traces show unrepresentative signal amplitudes?*

The example trace in Figure 1 was chosen for aesthetic reasons and is not a good representative of the average amplitudes. A new representative trace has been chosen that closely matches the average amplitude of 0.67 dF/F for that experiment.

*- I still do not understand the arguments surrounding Figure 4. If MK-801 were to block all evoked responses entirely, all data points would lie on the line from x,y = 0,0 to 1,-1 simply because the maximum -∆Rp equals Rp before. The figure shows a rather linear slope of -0.8 suggesting 80% inhibition of Rp whatever it was before MK-801 was added and I do not recognize any relation between blocking efficiency with Rp. ∆Rp may therefore also not be a good choice as analyzed parameter in panel A. Maybe the relative change ∆Rp/Rp(before) might be more instructive.*

We thank the reviewer for pointing this out, we now reconstructed these plots and changed the y axis to “percent change in R_p_ after MK-801” to illustrate that we are measuring the amount of MK-801 induced block by comparing the R_p_ before and after MK-801 treatment (∆Rp/Rp(before) plotted as a percentage). In these experiments, the efficiency of MK-801 is static, but the availability of its binding site depends on how many times the channel is opened. The meaning of these panels is to illustrate that the amount of MK-801 induced block of NMDARs that respond to evoked glutamate does not correlate with the amount of spontaneous activity during MK-801 application (panel 4A). Channels that open during the spontaneous-only period of MK-801 treatment will be blocked and unavailable to participate in the second round of evoked responses if they were to be activated by evoked glutamate release. However, this does not mean that MK-801 is not blocking receptors that receive evoked glutamate in a use dependent manner, because the amount of block is strongly correlated with the number of responses to stimulation (panel 4B). In other words, we observe that within a single synapse it is possible to subdivide the populations of NMDARs that receive evoked glutamate or spontaneous glutamate by preferentially blocking one with MK-801 and not the other. This is contrary to the prevalent assumption that the postsynaptic population of NMDARs form a homogeneous pool of receptors that are equally likely to receive glutamate from a spontaneous vesicle fusion event or an evoked vesicle fusion event.

To better explain this concept, the Results section has been edited to include the following:

“During stimulation there was a decrease in amplitude with every subsequent response in the MK-801 treated cells and no single ROI responded more than 8 times with MK-801 in the bath solution (Figure 3). This decrease in amplitude is accompanied by a decrease in R_p_ after MK-801 treatment that preferentially affects high responding ROIs which is consistent with its usage-dependent mechanism of action (Figure 3). Channels that open during the spontaneous-only period of MK-801 treatment will be blocked and unavailable to participate in the second round of evoked responses if they were to bind glutamate.

Comparing the spontaneous event rate measured *before* the addition of MK-801 against the change in evoked R_p_ after MK-801 treatment shows little correlation between the two parameters (Figure 4). If NMDARs were equally likely to receive neurotransmitter from a spontaneous vesicle fusion as an evoked vesicle fusion, we would expect that the ROIs with the highest rate of spontaneous activity would exhibit the largest amount of MK-801 induced blockade when measuring evoked responses. However, we find that the rate of spontaneous activity is a poor predictor of the amount of decrease in evoked responses. Comparing the initial R_p_ against the MK-801 induced change reveals that ROIs with the highest initial R_p_ see the largest decrease in R_p_ after treatment. These findings are consistent with the blockade of NMDA receptors responding to stimulation during MK-801 treatment being due to evoked glutamate release alone, with little contribution from spontaneous neurotransmission.”

[Editors' note: further revisions were requested prior to acceptance, as described below.]

*There remains some confusion about Figure 4. One reviewer points out that the fact that the data fall on a line suggest that MK801 blocks about 80% of the channels regardless of initial Rp; the percent block necessarily decreases as Rp decreases because there are fewer channels activated that can be blocked. Another reviewer suggests that this is actually the point: MK801 blocking efficiency may be constant at ~80% but the fact that it is predictable from evoked Rp but not spontaneous rate supports the separation of the groups. Since the reviewers had such difficulty working this out, a clarification in the text would be helpful. Also, Figure 4 is not called in the text. Please add a reference to this plot at the appropriate place in the manuscript. Finally, a minor question remains regarding the relationship between the amplitudes of the example signal in Figure 3 and the mean summary data in Figure 3 (see reviewer comment below). Please indicate whether the traces shown unusually large or whether there is an error in the scale bar.*

We now further clarified these remaining issues in the text and the legend of Figure 3.

*Additional comments from the reviewers are included below so that you can see the nature of the confusion of the reviewers.*

*The response to the following point is not clear to me:*

*"I still do not understand the arguments surrounding Figure 4. If MK-801 were to block all evoked responses entirely, all data points would lie on the line from x,y = 0,0 to 1,-1 simply because the maximum -ΔRp equals Rp before. The figure shows a rather linear slope of -0.8 suggesting 80% inhibition of Rp whatever it was before MK-801 was added and I do not recognize any relation between blocking efficiency with Rp. ΔRp may therefore also not be a good choice as analyzed parameter in panel A. Maybe the relative change ΔRp/Rp(before) might be more instructive."*

The y axis shows “percent change in R_p_ after MK-801” to illustrate that we are measuring the amount of MK-801 induced block by comparing the R_p_ before and after MK-801 treatment (∆Rp/Rp(before) plotted as a percentage). 80% inhibition of Rp only applies to high Rp values (0.8 –1) whereas lower Rp (e.g. around 0.2) values only see 10-20% block. In a way, this plot validates the classical assumption on the nature of MK-801 block at the single synapse level “higher the Rp (a proxy for release probability), larger the percentage of block by MK-801”. However, this increase in block – while proportional to initial Rp – is not correlated to the amount of spontaneous events seen prior to MK-801 application (panel 4A). This result indicates that a classical measure of release probability does not show a correlation between spontaneous response rate and evoked response rate at the single synapse level.

The section is expanded to further clarify this issue as follows:

“In contrast, comparing the initial Rp against the MK-801 induced change reveals that ROIs with the highest initial Rp see the largest decrease in Rp after treatment (Figure 4). These findings are consistent with the blockade of NMDA receptors responding to stimulation during MK-801 treatment being due to evoked glutamate release alone, with little contribution from spontaneous neurotransmission. In other words, the plot shown in Figure 4 validates the classical assumption on the nature of MK-801 block at the single synapse level “higher the Rp (a proxy for release probability), larger the percentage of block by MK-801”. However, this increase in block – while proportional to initial Rp – is not correlated to the amount of spontaneous events seen prior to MK-801 application (Figure 4). This result indicates that a classical measure of release probability does not show a correlation between spontaneous response rate and evoked response rate at the single synapse level.”

*"The ΔF/F0 values in Figure 3 seem almost an order of magnitude smaller than in the example traces of panel A. Is the scale bar in A wrong or the values in D or do the traces show unrepresentative signal amplitudes?"*

We apologize for this continuing confusion. In the sample traces we tried to convey the impact of MK-801 block on single events therefore we opted to choose examples that are on the larger the side of the average values (especially those after MK-801 application) to better convey the impact of MK-801 block on single events.

The following phrase is now added to figure legend 3A:

“Please note that sample traces were selected among examples that are on the larger the side of the average values (especially those after MK-801 application) for better visualization of the impact of MK-801 block on single events”